# Multiple Potential Plant Growth Promotion Activities of Endemic *Streptomyces* spp. from Moroccan Sugar Beet Fields with Their Inhibitory Activities against *Fusarium* spp.

**DOI:** 10.3390/microorganisms9071429

**Published:** 2021-07-02

**Authors:** Yassine Aallam, Bouchra El Maliki, Driss Dhiba, Sanaa Lemriss, Amal Souiri, Abdelmajid Haddioui, Mika Tarkka, Hanane Hamdali

**Affiliations:** 1Laboratory of Biotechnology and Valorization of Plant Genetic Resources, Faculty of Sciences and Technology, University of Sultan Moulay Slimane, P.O. 523, Beni-Mellal 23000, Morocco; yassine.aallam@gmail.com (Y.A.); slemriss@lram-fgr.ma (S.L.); ahaddioui@yahoo.fr (A.H.); 2Faculty of Medecine and Pharmacy, University Hassan II, Casablanca 20250, Morocco; b.elmaliki@gmail.com; 3International Water Research Institute (IWRI), Moulay Rachid, University Mohammed 6 Polytechnic (UM6P), Ben Guerir 43150, Morocco; driss.dhiba@um6p.ma; 4Laboratory of Research and Medical Analysis of Gendarmerie Royale, Department of Biosafety PCL3, Rabat 10090, Morocco; asouiri@lram-fgr.ma; 5UFZ—Helmholtz-Centre for Environmental Research, Soil Ecology Department, Theodor-Lieser-Straße 4, 06120 Halle, Germany; mika.tarkka@ufz.de

**Keywords:** *Streptomyces* spp., potassium solubilization, orthoclase, biocontrol, root rot, sugar beet

## Abstract

The characterized 10 *Streptomyces* isolates were previously selected by their abilities to solubilize phosphates. To investigate whether these isolates represent multifaceted plant growth-promoting rhizobacteria (PGPR), their potassium-solubilizing, auxin-producing and inhibitory activities were determined. The 10 *Streptomyces* spp. yielded a variable biomass in the presence of insoluble orthoclase as the sole potassium (K) source, indicating that they were able to extract different amounts of K from this source for their own growth. Three strains (AZ, AYD and DE2) released soluble K from insoluble orthoclase in large amounts into the culture broth. The production levels ranged from 125.4 mg/L to 216.6 mg/L after 5 days of culture. Only two strains, *Streptomyces enissocaesilis* (BYC) and *S*. *tunisiensis* (AI), released a larger amount of soluble K from orthoclase and yielded much more biomass. This indicated that the rate of K released from this insoluble orthoclase exceeded its consumption rate for bacterial growth and that some strains solubilized K more efficiently than others. The results also suggest that the K solubilization process of AZ, AYD and DE2 strains, the most efficient K-solubilizing strains, involves a slight acidification of the medium. Furthermore, these 10 *Streptomyces* spp. were able to secrete indole acetic acid (IAA) in broth medium and ranged from 7.9 ± 0.1 µg/mL to 122.3 ± 0.1 µg/mL. The results of the antibiosis test proved the potential of the 10 tested strains to limit the growth of fungi and bacteria. In dual culture, *S*. *bellus* (AYD) had highest inhibitory effect against the three identified fungal causal agents of root rot of sugar beet: *Fusarium equiseti* and two *F*. *fujikuroi* at 55, 43 and 36%, respectively. *Streptomyces enissocaesilis* (BYC), *S*. *bellus* (AYD) and *S*. *saprophyticus* (DE2) exhibited higher multifaceted PGPR with their potassium-solubilizing, auxin-producing and inhibitory activities, which could be expected to lead to effectiveness in field trials of sugar beet.

## 1. Introduction

One of the most commonly cultivated plants used in the sugar industry is sugar beet (*Beta vulgaris* L.). Production of sugar from this crop is the second largest in the world after production from sugar cane [1], but sugar beet is more resilient, coping with less favorable environmental conditions than sugar cane [2]. In Morocco, sugar beet occupies an area of 57,171 hectares with a production of 3.7 million tons in 2019 [3]. The Beni Mellal- Khenifra region (east of Casablanca), located in a vast agricultural plain in Morocco, represents 26% of the national production with an area of 15,000 hectares [4], and contributes 20.5% and 21.2% to the national area and production of sugar beet in Morocco, respectively [5].

Sugar beet requires balanced fertilization by the three macronutrients, nitrogen (N), phosphorus (P) and potassium (K). Potassium is one of the most important elements for sugar beet nutrition and plays a significant role in enzyme activation, charge balance, osmoregulation and reduction in the negative effects of drought stress [6,7]. In sugar beet, K is also involved in biosynthesis and transfer of sucrose to storage roots [8]. Hanafy et al. [9] found that the application of K significantly improved root yield, length, diameter and fresh root weight of sugar beet. Additionally, many studies have reported that K and N enhance the biosynthesis of organic metabolites which increase the yield and quality of sugar beet [10]. In soil, K is a mineral element which represents a minor soluble part, ranging from 1.00 to 1.5 × 10^−3^ mmol/L, and is thus unavailable for plant uptake [11]. In soil, potassium (98%) exists in the form of silicate minerals (orthoclase, feldspars, muscovite, microline, etc.) [12].

In contrast to the well-documented action of phosphate-solubilizing and nitrogen-fixing microorganisms in plant P and N nutrition [13,14], the importance of microbial activity on plant K nutrition is less well understood. It has been shown that bacteria and mycorrhizal fungi do solubilize mineral K by various mechanisms such as production of alkaline substances including ammonia and other metabolites upon death of microbial cells and subsequent proteolysis [15,16]. K-solubilizing bacteria (KSB) have attracted more attention because of their ability to increase plant growth and yield [17]. However, very few studies have demonstrated these abilities of KSB [18,19,20,21,22,23]. To date and to the best of our knowledge, this is the first time that such a study has been undertaken, especially on the endemic Actinobacteria from sugar beet rhizospheres in the Beni-Mellal region with these PGPR criteria.

As in all plants, sugar beets are parasitized by a number of fungi that are able to attack one or more of their organs, reducing their growth and the reserves accumulated in roots. Rhizoctonia solani represents one of the major constraints in the production of sugar beets around the world [24] and is the main cause of various plant diseases, especially root rot [25]. *Fusarium* sp. is another damaging agent of sugar beet, causing root rot [26,27,28]. Plants infected with *Fusarium* spp. show early symptoms: the color of the leaves turns yellow and they die, the petioles also wilt around the crown and the root yield and the percentage of sucrose are reduced [29]. Moreover, *Rhizoctonia crocorum*, *Phoma betae*, *Aphanomyces cochlioides*, *Macrophomina phaeseolina*, *Pythium aphanidermatum*, *Phytophthora drechsleri*, *Sclerotium rolfsii* and *Rhizopus stolonifer* are also causative agents of root rot of sugar beets [30,31].

Actinobacteria are known by the production of phytohormones such as ethylene, gibberellic acid, indole-3-acetic acid (IAA), cytokines and ACC deaminases [32,33,34,35]. Moreover, Actinomycetes can have antimicrobial activities and effects against phytopathogens [36,37,38]. From the 10,000 antimicrobial molecules produced by Actinomycetes, more than 50% were isolated from Streptomycetes [39].

Therefore, this research aimed to assess if the 10 strongly P-solubilizing strains of Streptomyces of the sugar beet rhizosphere in Beni-Mellal [40] possess multifaceted PGPR activities, and tested their abilities to solubilize K from insoluble orthoclase, produce auxins and inhibit sugar beet root pathogens in laboratory conditions. Through these experiments, we expected to identify the most effective strains of the collection for future field trials.

## 2. Materials and Methods

### 2.1. Actinomycete Strains

Ten selected Actinomycete strains used in this study were isolated from three different sugar beet rhizospheres that were previously selected for their abilities to solubilize insoluble phosphate forms (a Moroccan natural rock phosphate and a synthetic tricalcium phosphate) using the SMM medium and to produce siderophores [40]: AYD, AZ, AV and BX related to *Streptomyces bellus*; BYC: *Streptomyces enissocaesilis*; AI: *Streptomyces tunisiensis*; BP: *Streptomyces coerulescens*; CYM: *Streptomyces cyaneofuscatus*; DE1: *Streptomyces bellus* and DE2: *Streptomyces saprophyticus*. Spores of the selected Actinomycete isolates were stored in 20% sterile glycerol at −20 °C for the next tests.

### 2.2. Actinomycete Abilities to Use Orthoclase as Sole Potassium Source

Potassium solubilization by the 10 Actinomycete strains was carried out on modified Aleksandrov agar medium (AMM) [41], containing 5.0 g/L glucose, 0.5 g/L MgSO_4_·7H_2_O, 0.1 g/L CaCO_3_, 0.006 g/L FeCl_3_, 2.0 g/L Na_2_HPO_4_ and orthoclase powder (2.0 g/L) [42] as a unique K source, or on AMM containing soluble K_2_HPO_4_ (0.5 g/L) as a positive control or no K source as a negative control. After plating, the agar plates were incubated for 21 days at 28 °C in order to allow growth of the slow-growing Actinomycetes.

### 2.3. Quantitative Estimation of the Amount of Solubilized Potassium by the Selected Actinomycete Strains

Three different tests with 10^6^ spores/mL of each of the ten selected Actinomycete strains were inoculated in 250 mL Erlenmeyer flasks containing 50 mL of liquid AMM with 2 g/L of orthoclase as the sole natural K source were carried out in triplicate, and the cultures were grown for 5 days at 28 ± 2 °C on a rotary shaker (180 g/min) [41]. Every day, a 1 mL aliquot of each culture was taken and centrifuged at 10,000× *g* for 10 min, and dry biomass and the pH of the supernatant were determined. The supernatant was analyzed for soluble K content by inductively coupled plasma atomic emission spectroscopy according to Liu et al. [43]. Similar measures were carried out in non-inoculated flasks incubated in the same conditions.

### 2.4. Assay for Indole Acetic Acid (IAA) Production

The production of IAA was evaluated on liquid Luria–Bertani (LB) medium supplemented with L-tryptophan (Merck, Darmstadt, Germany) (1 g/L) [44]. The growth medium used contained tryptone (Merck) (10 g); yeast extract (Merck) (5 g); NaCl (5 g) in 1 L of deionized water. This medium was sterilized at 121 °C for 20 min. Sterilized tubes containing 10 mL of the medium were inoculated with 20 μL of actinobacterial cultures (10^7^ cfu/mL) and incubated at 28 °C for 72 h with moderate stirring at 130 rpm/min. The determination of the IAA concentration was carried out by the addition of 2 mL of Salkowski’s reagent (2% (*w*/*v*) 0.5 M FeCl_3_ in 35% perchloric acid) to 1 mL of the culture supernatant previously centrifuged for 10 min at 10,000× *g*. A pink color indicated the presence of IAA. The IAA concentration produced was estimated by measuring the OD at 530 nm after 30 min of incubation in the dark [45]. The concentration of IAA produced by each Actinomycete strain was determined by the generation of a standard curve.

### 2.5. Isolation of Fungi from Symptomatic Sugar Beet Roots

Filamentous fungi were collected during a sugar beet harvesting campaign in June 2020. The contaminated sugar beet site was located 25 km from Beni Mellal city (32°29′34.2″ N 6°10′56.1″ W, center of Morocco). This infected site has a warm Mediterranean climate with a temperature ranging between 1.1 and 40 °C and a mean annual rainfall generally between 350 and 650 mm/year [46]. In sterile plastic bags, roots with symptoms of damage and fungi infection were harvested and isolated aseptically (Figure 1).

According to the modified protocol of Chenaoui et al. [47], the surface soil of contaminated sugar beet root was removed. To cultivate rhizoplane fungi that may be associated with the root symptoms and interact with rhizosphere Streptomycetes, the mycelium on the root surface was scraped off with a sterile platinum loop and plated on potato dextrose agar (PDA) medium (Merck, France). The PDA medium used contained potato infusion (4 g), dextrose (20 g) and agar (17 g) in 1L of deionized water at a final pH of 5.6 ± 0.2. Plates were incubated in the dark at 26 ± 2 °C for 7–8 days and checked every day. Hyphae were purified and preserved on PDA medium for further study.

### 2.6. Molecular Determination of Isolated Fungi (DNA Extraction and ITS Amplification)

Preliminary identification of the three selected fungi (CHAMP1, CHAMP2 and CHAMP3) was performed by microscopic observation of the plates with lacto phenol blue, direct examination with 20% KOH Chinese ink and by description of the macroscopic features of colonies after a period of incubation of eight to ten days at 25 °C.

For molecular identification, fungal DNA was isolated from pure cultures on Sabouraud agar medium of three selected isolates. Approximately 1 cm^2^ of mycelium was collected and added to an Eppendorf tube containing 1 mL of distilled water and 12 to 15 glass beads with a diameter of 3 mm (Merck). After vortexing at maximum speed for 2 min, 300 µL of the suspension containing fragmented mycelium was used for automated extraction of DNA with the Maxwell^®^ RSC Instrument (Promega, Madison, WI, USA) and the Maxwell RSC PureFood GMO and Authentication Kit (Promega) according to the manufacturer’s instructions. The nuclear ribosomal ITS1-5.8S-ITS2 region was amplified with the universal primers ITS1 (5′-CCGTAGGTGAACCTGCGG-3′) and ITS4 (5′-TCCTCCGCTTATTGATATGC-3′) [48]. The nuclear rDNA segment corresponding to the D2 region of the fungal 28S ribosomal RNA (28S-D2 rDNA) was amplified with the primers NL1 (5′-GCATATCAATAAGCGGAGGAAAAG-3′) and NL4 (5′-GGTCCGTGTTTCAAGACGG-3′) [49]. The 18S ribosomal RNA region was amplified with the primers FR1 (5′-ANC CAT TCA ATC GGT ANT-3′) and FF390 (5′-CGA TAA CGA ACG AGA CCT-3‘) [48].

Amplifications were carried out in a 25 mL reaction mixture containing 5 µL of PuRe Taq Ready-To-Go PCR Mix (GE Healthcare, Little Chalfont, UK), 4.5 pmoL of each primer, 9.1 µL of water and 50 ng of DNA. PCR conditions were as follows: after initial denaturation (95 °C for 3 min), 45 cycles of 95 °C for 1 min, 54 °C for 1 min s and 72 °C for 2 min, followed by a final extension (7 min, 72 °C). All amplifications were carried out using a GeneAmp PCR 9700 System (Applied Biosystems). Negative controls were included with no addition of template DNA. PCR products were visualized on a 2% (*w*/*v*) agarose gel stained with ethidium bromide. The nuclear ribosomal ITS1-5.8S-ITS2 region, 28S ribosomal RNA region and 18S ribosomal RNA region were sequenced bidirectionally with primers ITS1 and ITS4, NL1 and NL4, and FR1 and FF390, respectively.

Sequences similarities were performed against corresponding sequences of *Fusarium* species using GenBank through Nucleotide BLAST (http://www.ncbi.nlm.nih.gov/BLAST/ (accessed on 3 April 2021)). An unrooted phylogenetic tree was inferred using the neighbor-joining method [50]. The percentage of replicate trees in which the associated taxa clustered together in the bootstrap test (1000 replicates) is shown next to the branches [51]. The evolutionary distances were computed using the Kimura 2-parameter method [52] and are in the units of the number of base substitutions per site. Evolutionary analyses were conducted in MEGA X [53].

### 2.7. In Vitro Antagonistic Activity Assays

The plate diffusion method was used to assess the wide-spectrum antimicrobial activity of the 10 selected strains [54] against the following Gram-negative bacteria: *Escherichia coli* CCMM/B4 and *Pseudomonas aeruginosa* CCMM/B90, *Klebsiella pneumonia* E40, *Salmonella* sp. CCMM/B17; the following Gram-positive bacteria: *Bacillus subtilis* ATCC 9524, *Staphylococcus aureus* CCMM/B2, *Streptococcus* sp. CCMM/B24; and the yeast *Candida albicans* CCMM/L11 (all strains from the collection of BVRP Laboratory, USMS Beni Mellal) and three fungi isolated from the infected field of sugar beet rhizospheres, Beni Mellal, Morocco, as described above. These 10 selected *Streptomyces* spp. were grown on solid SMM [55] for 14 days and then three disks (diameter 10 mm) were cut out and placed on lawns grown for 48 h of the different microorganisms on nutrient agar (Difco, Sparks, MD, USA) for bacteria and yeast. Plates were first stored at −4 °C for at least 2 h to allow the diffusion of any substances produced, and then incubated at 28 °C. Sizes of the inhibition zones were determined after 24 h of incubation for bacteria and yeast. Controls involved the use of sterile agar plugs. Three replicates were performed for each isolate in each microorganism test.

For fungi, we followed the technique of dual culture from Karimi et al. [56], which was slightly modified. Briefly, disks of 10 mm in diameter from 5-day-old cultures of fungi were placed in the center of Sabouraud agar plates and the disks of tested Actinomycete strains were placed on either side of Sabouraud plates at a distance of 2.5 cm. After 5 days of incubation at 28 °C, the diameter of propagation of the fungi was recorded and the percentage of inhibition (PI) was calculated: PI = [(T − S)/T] × 100; where T represents the mycelial colony radius (mm) of a control culture and S represents the mycelial colony radius of the fungus in the presence of the Streptomyces strain [40].

### 2.8. Statistical Analysis and Detection of the Most Promising Strains for Simultaneous Biofertilizer and Biocontrol Applications

All experimental results were subjected to analysis of variance (ANOVA) using SPSS statistical software, version 23.0 (IBM, New York, NY, USA). The results were expressed as mean ± SD. The means were compared using the least significant difference (LSD) test. *p* < 0.05 indicates significant differences. To select the most promising Streptomycete isolates for combined biofertilizer and biocontrol applications, their relative activities in rock phosphate and tricalcium phosphate and of potassium solubilization, antifungal activity against *Fusarium* spp., antibacterial activity and production of IAA and siderophores were cross-compared between the strains and used to grade them from 1 (lowest) to 10 (highest activity). In order to achieve the aim of selecting combined biofertilizer and biocontrol agents, the rating values for rock phosphate and potassium solubilization, IAA production and Fusarium inhibition were weighted by 2 (doubled).

## 3. Results

### 3.1. Growth Kinetics of the Selected Actinomycete Strains in AMM + Orthoclase

The ten tested Actinomycete strains exhibit clear development on solid AMM with orthoclase as the sole potassium source. Figure 2 shows that the growth kinetics of most strains was variable from strain to strain. This indicated that the strains were able to assimilate variable amounts of potassium from these insoluble K sources, with a different efficiency, and use it for their own growth. The only exceptions were the strains BYC and AI that showed a better growth on AMM + orthoclase. In mainly AMM + orthoclase, all strains yielded a lower biomass at day 5 than at day 4, suggesting cell lysis. However, the biomass yield was not the same for all strains and the strains could be grouped into three classes: class I with biomass yield ≥ 600 µg/mL (BYC and AI), class II with biomass yield between 300 and 600 µg/mL (AZ, DE1, BP, CYM, DE2, AV and AYD) and class III with biomass yield below 200 µg/mL (BX strain) (Figure 2).

### 3.2. Estimation of the Amount Soluble Potassium Released from Orthoclase by the Selected Actinomycete Strains

The concentration of soluble K in the supernatant of all 10 selected Streptomyces strains from the sugar beet rhizospheres was assessed (Figure 3).

All strains were able to release potassium, except for the DE1 strain that likely used it for its own growth. The amount of soluble K released greatly varied from strain to strain and ranged from 3.8 mg/L to 216.6 mg/L. Three strains (AZ, AYD and DE2) released a large amount of potassium in the AMM supernatants (>100 mg/L) (Figure 3). The presence of large amounts of K in the supernatant of these strains in the presence of orthoclase simply indicated that the rate of K released from orthoclase exceeded its rate of consumption for bacterial growth. As expected, strains with the lowest biomass yields (Figure 2) solubilized the smallest amounts of potassium (Figure 3).

Strains AZ and AYD (class II) released the maximal soluble K concentration (216.6 mg/L and 155.6 mg/L, respectively), whereas strains BYC and AI (class I) released approximately 2- to 3-fold smaller amounts of soluble K from orthoclase (61 mg/L and 35.8 mg/L, respectively) (Figure 3).

The strain BYC (class I) is of special interest since it was able to release a large amount of K from orthoclase (61 mg/L) and its biomass yield was the highest (Figure 3). The strains AYD and DE2 (class II) released over 2-fold more soluble K from orthoclase (155.6 mg/L and 125.4 mg/L, respectively) than the BYC strain (Figure 3). The biomass yield of the strain AI was 14% lower than that of the strain BYC in similar culture conditions. Interestingly, the strains DE1 (class II) and BX (class III) did not increase the amount of soluble potassium in the culture medium. This suggested that the K-solubilizing ability of these strains was less efficient than that of the others.

### 3.3. PH Evolution of the Growth Medium

The pH of the growth medium of all strains in orthoclase was between 7.2 and 6.2 at day 1 and stabilized afterwards in most cases at approximately pH 6.2, except for AI, CYM and AYD strains (Figure 4). This indicated that the solubilization process might involve the excretion of organic acids. The pH of the medium decreased below 6 for the AI strain from day 3 (Figure 4). The AYD strain released the largest amount of K (155.6 mg/L) whereas AI released a fair amount of K from orthoclase (35.8 mg/L) and CYM released a rather small amount (5.15 mg/L) (Figure 3). The pH of the medium of the seven remaining strains (AZ, AYD, DE2, BYC, AV, BP and BX) was similar and stabilized at pH 6 (Figure 4), suggesting their abilities to solubilize K by organic acid production.

### 3.4. Indole Acetic Acid (IAA) Production

The results of IAA production of the 10 selected *Streptomyces* spp. are shown in Figure 4. IAA production was detected in all strains in significantly different amounts, ranging from 7.9 to 22.3 μg/mL. The highest concentration was produced by the CYM and BYC strains (22.3 ± 0.1 µg/mL and 19.8 ± 0.1 µg/mL) followed by AV (16.4 ± 0.8 µg/mL), AYD 16.3 ± 0.1 µg/mL) and DE2 (16.2 ± 0.9 µg/mL). The lowest values, 7.9 ± 0.1 µg/mL and 8.8 ± 1.0 µg/mL, were obtained by BP and AZ, respectively (Figure 5).

### 3.5. Identification of the Isolated Fungi from the Field

The result of preliminary identification shows that the three selected isolates (CHAMP1, CHAMP2 and CHAMP3) belong to the genus *Fusarium*. Macroscopically, most colonies were characterized by their cottony appearance, salmon pigmentation and purple, lilac, light brown, yellow or gray color. Microscopically, colonies were characterized by hyaline hyphae, septate and filamentous; septate macroconidia, fusiform with characteristic appearance of alantoespora, oval microconidia and, in some cases, chlamydoconidia of thick walls.

The fungi sequences of three selected isolates were analyzed using BLAST (http://www.ncbi.nlm.nih.gov/BLAST/ (accessed on 3 April 2021)). They belonged to the *Fusarium* genus, bearing an identity of at least 99% and confirmed our result of preliminary identification.

Nucleotide sequences of the identified fungi were deposited in the GenBank Database (http://www.ncbi.nlm.nih.gov/GenBank/ (accessed on 3 April 2021)), and were assigned accession no. MW824656 (*Fusarium equiseti*), MW824923 (*Fusarium fujikuroi*) and MW825349 (*Fusarium fujikuroi*).

A small subunit ribosomal RNA gene, internal transcribed spacer 1 and 5.8S ribosomal RNA gene and internal transcribed spacer 2 sequences (513 nt) of 28 *Fusarium* species retrieved from GenBank, as well as those of our strains, were used for the construction of a phylogenic tree (Figure 6). Two fungi (CHAMP1 and CHAMP3) were closely related to *F. fujikuroi* and the isolate CHAMP2 was related to *F. equiseti*.

### 3.6. In Vitro Antagonistic Activity Assays

We evaluated the antagonistic activity of our 10 selected *Streptomyces* spp. against the three identified sugar beet rhizoplane fungi, pathogenic bacteria and yeast (Figure 7). All *Streptomyces* spp. showed an antagonistic activity at different inhibition percentages (PIs). PIs ranged from 6.9% (BX strain) to 55.4% (AYD strain). The AYD strain exhibited a significantly higher inhibitory effect (PI) against the identified fungi, two *Fusarium fujikuroi* (CHAMP1) and *F. equiseti* (CHAMP2), at 55.4% ± 0.6 and 43.1% ± 0.2, respectively (Figure 7), whereas the lowest activity was shown by the AZ strain.

None of the 10 tested Streptomyces isolates showed activity against the Gram-positive *Bacillus subtilis* or *Streptococcus* sp., or Gram-negative *Pseudomonas aeruginosa*. Significant inhibition activity was shown by CYM, DE1 and AZ strains against *Staphylococcus aureus*. The 10 *Streptomyces* spp. were not able to limit growth of the Gram-negative bacterium *Escherichia coli*, except the AYD strain. Only the BYC and AYD strains had a significantly strong inhibitory activity against *Salmonella* sp. Except the AYD and AV strains, all the tested strains had inhibitory activity against *Klebsiella pneumonia*. The BYC, BP and DE1 strains produced the largest inhibition zones. Nine out of the ten *Streptomyces* sp. (except the BX strain) inhibited growth of *Candida albicans*, and the DE2, CYM and DE1 strains produced the largest inhibition zones (Table 1).

Finally, the results were combined together to identify the multifaceted strains with the highest potential for sugar beet growth promotion in order to use them for further experiments in field trials (Table 2).

## 4. Discussion

In our previous research [40], we successfully isolated 10 *Streptomyces* spp. and revealed their phosphate solubilization capacities and siderophore production. In this study, we investigated whether these *Streptomyces* spp. represent multifaceted PGP by testing their abilities to solubilize potassium from orthoclase as the sole K source and to produce IAA and their inhibitory activities against *Fusarium* sp., the causal agent of sugar beet root rot in the Beni Mellal region (Morocco).

More than 90% of K exists as an insoluble rock form in soils, and the amount of soluble K is very small [57]. It has become a necessity to isolate and screen K-solubilizing microorganisms in order to improve the amount of soluble K in soil and to meet the requirements of growing plants. Thus, the results of this study showed that the 10 *Streptomyces* spp. were able to solubilize K using AMM with orthoclase as the sole K source. Interestingly, in the AMM broth, the amount of soluble K ranged from 3.8 mg/L to 216.6 mg/L after 5 days of cultivation. Similarly, Han et al. [58] reported that *Streptomyces rochei* and *S*. *sundarbansensis* released amounts of K of 7.46 to 13.71 mg/L in the same conditions. Other studies showed that *Streptomyces alboviridis* P18, *S*. *griseorubens* BC3, *S*. *griseorubens* BC10 and *Nocardiopsis alba* BC11 had the ability to solubilize K in AMM and amounts of K ranged from 2.6 to 41.45 mg/L after 11 days of cultivation [23]. Reyes-Castillo et al. [59] reported that *Bacillus mucilaginosus* dissolved K from motmorillonite, kaolinite and K-feldspar, showing values of 90 to 140 mg/L after 7 days of cultivation.

Microorganisms have many mechanisms to solubilize K from a mineral source. Researchers have suggested that this solubilization might be either due to the excretion of protons, H+ or CO_2_ and organic acids, causing acidification of the external medium, or to the excretion of chelating substances (such as siderophores) that form stable complexes with potassium adsorbents (aluminum, iron and calcium) [57,60,61,62,63]. In this research, the pH of the growth medium of all strains in AMM broth with orthoclase was between 7.2 and 6.2 at day 1 and stabilized afterwards in most cases at approximately pH 6.2, except for the AI and CYM strains (Figure 2). This indicated that the solubilization process might involve the excretion of organic acids. Interestingly, AI yielded much better biomass (Figure 3) and was shown to excrete siderophores, but not the CYM strain [40], whereas BYC released a rather small amount (2-fold less, 61 mg/L). However, this strain produced siderophores and was selected as the best phosphate-solubilizing strain from both RP and TCP [40]. The purification and structural characterization of potentially novel siderophores and/or organic acids are in progress. Our results are in agreement with other investigations performed by Bagyalakshmi et al. [64], which reported that KSB prefer an acidic to neutral range of pH. Moreover, other researchers reported that the production of some different primary proteins and polysaccharides can also control the release of K from K-bearing minerals [65,66,67]. In contrast, Dhiman et al. [68] have observed that at pH 7, the strains show the best K solubilization activity in the presence of *Proteus mirabilis* MG738216.

Furthermore, the 10 selected *Streptomyces* spp., except the DE1 strain, had the ability to produce IAA (Figure 4). The CYM, BYC, AV, AYD and DE2 strains were the best IAA producers, with values of more than 15 µg/mL to 22.3 µg/mL. According to Djebaili et al. [32], the bacteria with the ability to produce more than 13.0 µg/mL have PGP activity. Similarly, *Streptomyces atrovirens* showed an interesting growth-promoting activity in groundnut, cotton and maize [69]. Several other studies have demonstrated the ability of Actinomycetes to secret IAA in several other plant rhizospheres [23,58].

Multiple potential PGP activities by the selected 10 *Streptomyces* spp. were performed by their antagonistic activity against three *Fusarium* sp. isolated and identified from the infected sugar beet rhizosphere (Figure 1 and Figure 5). In dual culture, S. *bellus* (AYD) had a higher inhibitory effect against the three fungal causal agents of root rot of sugar beet: *Fusarium equiseti* and two *F*. *fujikuroi* at 55%, 43% and 36%, respectively. Similarly, Getha et al. [70] demonstrated the ability of a *Streptomyces* sp. to inhibit the growth of *Fusarium* wilt of banana. Moreover, *Streptomyces bikiniensis* HD-087 isolated from Hulunbeier grassland soil show a strongly antagonistic activity against *Fusarium oxysporum* [71]. Streptomyces have also shown an ability to control *Fusarium* wilt of tomato in greenhouse conditions [72]. According to Yang et al. [73], 60% of antibiotics used in agriculture are produced by different *Streptomyces* species. To explain the biocontrol ability of Actinomycetes, many mechanisms have been suggested, such as secreting extracellular cell wall hydrolases [74], production of secondary metabolites [75], inducing plant resistance [76,77], competing for nutrients such as iron (siderophores) in the environment and by producing hydrolytic enzymes, especially chitinase, which degrades the crucial constituent of the fungal cell wall, chitin [78,79].

These results suggest that our selected species, *Streptomyces enissocaesilis* (BYC), *S*. *bellus* (AYD) and *S*. *saprophyticus* (DE2), exhibit the highest multifaceted PGPR activities, and they may represent efficient biofertilizers and biocontrol agents (Table 2). Further studies on the effectiveness of their use in controlled pot experiments and in the field are in progress. The use of these endemic selected *Streptomyces* spp. as both P and K solubilizers, in addition to their biological control ability, could be considered as an innovative pathway to sustainable and ecofriendly agriculture.

## Figures and Tables

**Figure 1 microorganisms-09-01429-f001:**
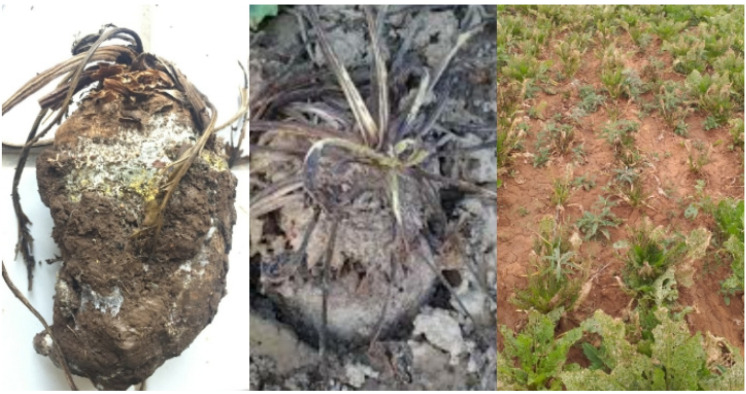
Symptomatic sugar beet roots and field with the isolated *Fusarium* spp. causal agent of root rot disease in Beni Mellal, Morocco.

**Figure 2 microorganisms-09-01429-f002:**
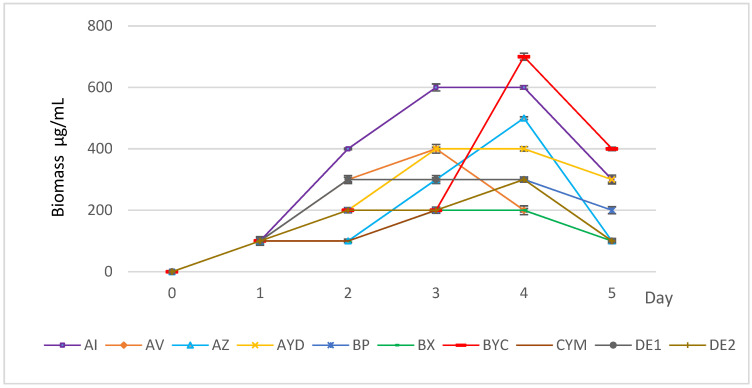
Evolution of the biomass of the selected Actinomycete strains grown in the liquid AMM + orthoclase. Error bars represent standard deviations of the mean values of the results of three independent culture replicates.

**Figure 3 microorganisms-09-01429-f003:**
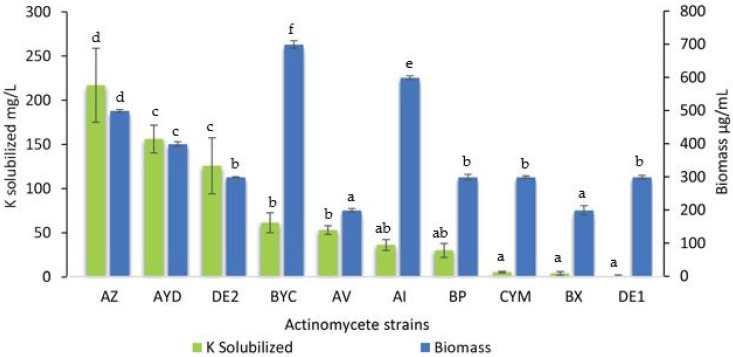
Concentration of soluble potassium released (mg/L) from orthoclase in the supernatant of cultures of the ten selected Actinomycete strains (green bars) grown for five days in AMM containing 2 g/L orthoclase with values subtracted from those of the non-inoculated flasks. The maximal biomass yield in µg/mL of the 10 selected Actinomycete strains is shown in the histogram (blue bars). Error bars represent standard deviations of the mean values of the results of three independent culture replicates. Different lowercase letters above bars show significant differences between treatments at *p* ≤ 0.05.

**Figure 4 microorganisms-09-01429-f004:**
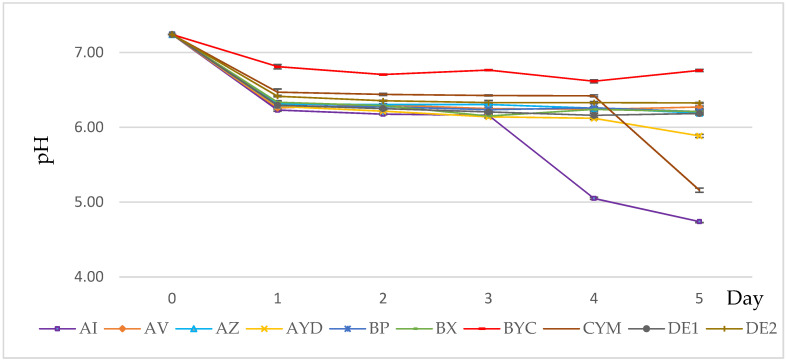
Evolution of the pH of the medium of the selected Actinomycete strains grown in AMM + orthoclase. Error bars represent standard deviations of the mean values of the results of three independent culture replicates.

**Figure 5 microorganisms-09-01429-f005:**
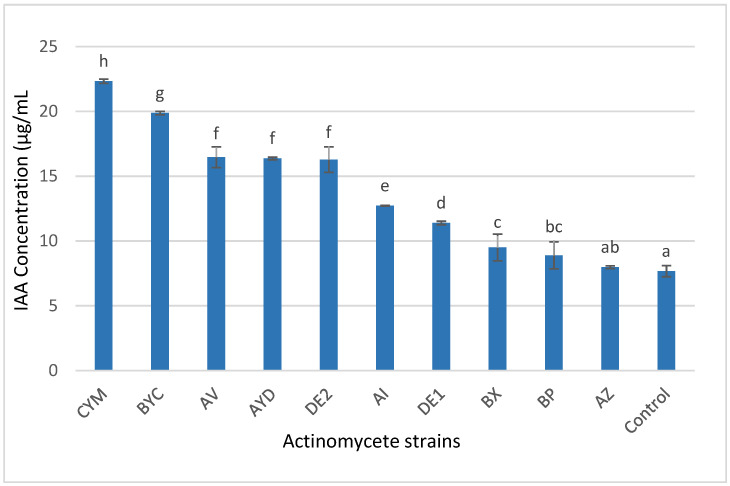
Concentration of indole acetic acid released (µg/mL) in the non-inoculated flasks (control) and in the supernatant of cultures of the ten selected Actinomycete strains grown for 72 h in liquid LB medium containing L-tryptophan (1 g/L). After addition of 2 mL of Salkowski’s reagent, the formation of the pink color indicated the presence of IAA and the IAA concentration produced was estimated by measuring the OD at 530 nm after 30 min of incubation in the dark. Error bars represent standard deviations of the mean values of the results of three independent culture replicates. Different lowercase letters above bars shows significant differences between treatments at *p* ≤ 0.05.

**Figure 6 microorganisms-09-01429-f006:**
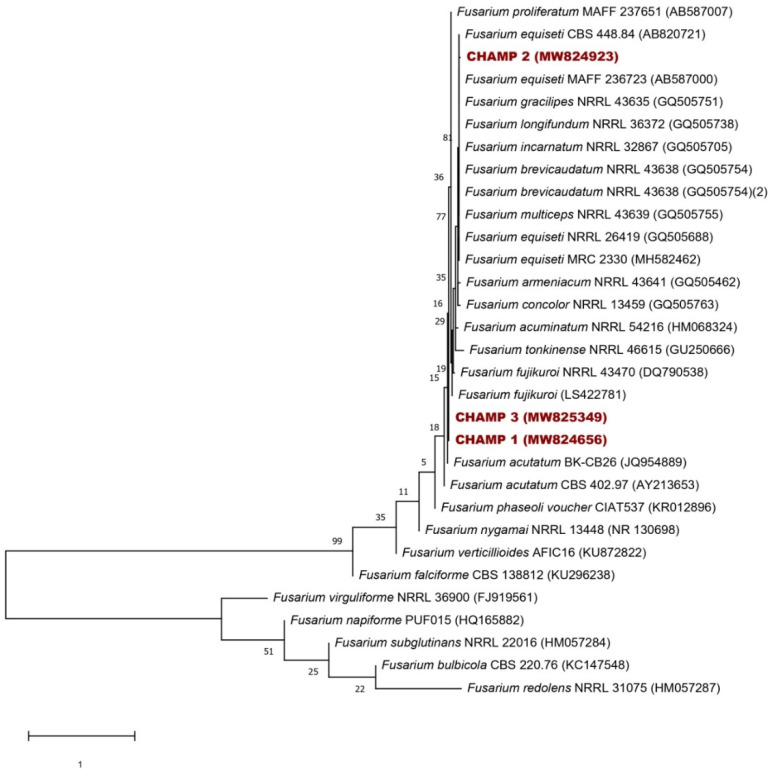
Neighbor-joining phylogenetic tree of the three isolated fungal strains (CHAMP 1, CHAMP 2 and CHAMP 3) and 28 *Fusarium* species based on small subunit ribosomal RNA gene, internal transcribed spacer 1 and 5.8S ribosomal RNA gene, and internal transcribed spacer 2 sequences (513 nt). Numbers at nodes indicate levels of bootstrap support (%) based on a neighbor-joining analysis of 1000 resampled datasets; only values > 50% are given. Accession numbers are given in parentheses. Bar marks one nucleotide substitution per site.

**Figure 7 microorganisms-09-01429-f007:**
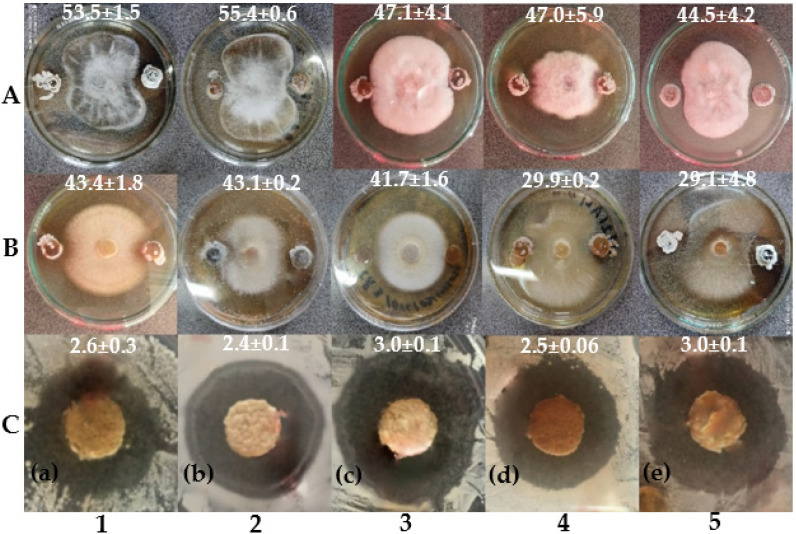
Performance comparison between the five most active *Streptomyces* spp. (1: DE2; 2: AYD; 3: BYC; 4: AI and 5: AV) against the three isolated phytopathogenic fungi from sugar beet rhizosphere in Beni Mellal, Morocco: (**A**) *Fusarium fujikuroi* (CHAMP1) and (**B**) *F. equiseti* (CHAMP2); (**C**) yeast: (**a**) *Candida albicans* CCMM/L11; pathogenic bacteria: (**b**) *Escherichia coli* CCMM/B4; (**c**) *Salmonella* sp. CCMM/B17; (**d**) *Klebsiella pneumonia* E40; (**e**) *Staphylococcus aureus* CCMM/B2.

**Table 1 microorganisms-09-01429-t001:** In vitro antagonistic activities of the ten Streptomyces strains against the three Fusarium isolates from diseased sugar beet roots from Beni Mellal, Morocco: Two *Fusarium fujikuroi* (CHAMP1; CHAMP3) and *F. equiseti* (CHAMP2); yeast: *Candida albicans* CCMM/L11; human pathogenic bacteria: *Escherichia coli* CCMM/B4; *Salmonella* sp. CCMM/B17; *Klebsiella pneumonia* E40; *Staphylococcus aureus* CCMM/B2; *Bacillus subtilis* ATCC 9524; *Streptococcus* sp. CCMM/B24; *Pseudomonas aeruginosa* CCMM/B90 (zone of inhibition ± SD cm) (percentage of inhibition ± SD mm). Different lowercase letters above bars show significant differences between treatments at *p* ≤ 0.05.

	Fungi			Bacteria Gram+	Bacteria Gram–			Yeast
*Streptomyces* Strains	CHAMP 1	CHAMP 2	CHAMP 3	*S. aureus* CCMM/B2	*Salmonella* sp. CCMM/B17	*K. pneumonia* E40	*E. coli* CCMM/B4	*C. albicans* CCMM/L11
	Percentage of Inhibition (PI ± SD mm)			Zone of Inhibition ± SD cm				
CYM	41.3 ± 8.8 d	35.4 ± 3.5 de	30.7 ± 5.7 b	2.8 ± 0.1 e	0	2.7 ± 0.1 bc	0	2.9 ± 0.1 d
BP	24.2 ± 1.2 c	8.2 ± 7.5 a	23.8 ± 5.3 ab	1.3 ± 0.0 a	0	3.0 ± 0.1 d	0	1.3 ± 0.1 a
DE1	20.4 ± 4.2 bc	13.7 ± 7.3 ab	23.0 ± 4.3 ab	2.5 ± 0.2 cd	0	2.8 ± 0.2 c	0	2.6 ± 0.1 c
AZ	7.6 ± 3.4 a	19.2 ± 11 abc	18.5 ± 2.6 ab	2.3 ± 0.2 c	0	2.2 ± 0.1 a	0	1.7 ± 0.5 ab
BX	12.7 ± 3.3 ab	7.6 ± 0.7 a	6.9 ± 0.4 a	2.2 ± 0.4 c	0	2.3 ± 0.1 a	0	2.3 ± 0.3 c

**Table 2 microorganisms-09-01429-t002:** The criteria score for screening of the 10 selected *Streptomyces* sp. with multifaceted PGP activities from sugar beet rhizospheres in Beni Mellal, Morocco. The numbers in the table represent a score out of 10, assigned to each strain according to its solubilization capacity, antimicrobial activity and the rate of IAA and siderophore release.

Streptomyces Strains	Solubilization of RP **^,^*^ψ^	Solubilization of TCP *	Potassium Solubilization ^ψ^	Antifungal Activity ^ψ^	Antibacterial Activity	Production of IAA ^ψ^	Production of Siderophores *	Final Score
BYC	20	9	14	16	4	18	9	90
AYD	16	8	18	20	3	14	8	87
DE2	10	10	16	18	7	12	10	83
AZ	14	7	20	4	6	2	7	60
AI	12	6	10	14	1	10	6	59
CYM	8	2	6	10	10	20	2	58
AV	4	4	12	12	5	16	4	57
BP	18	5	8	8	9	4	5	57
DE1	6	3	2	6	8	8	3	36
BX	2	1	4	2	2	6	1	18

* Aallam et al. [40], ^ψ^ criteria score doubled.

## Data Availability

The data that support the findings of this study are available from the corresponding author upon reasonable request.

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
