# Peer review of "Multiple Potential Plant Growth Promotion Activities of Endemic Streptomyces spp. from Moroccan Sugar Beet Fields with Their Inhibitory Activities against Fusarium spp."

_microorganisms, 2021, doi:10.3390/microorganisms9071429_

Round 1

Reviewer 1 Report

The manuscript is dedicated to the investigation of actinomycetes of genus Streptomyces as promising Plant Growth-Promoting Rhizobacteria (PGPR) for agricultural applications. The article presents the results of a comprehensive study of the activity of Streptomyces strains according to various parameters: solubilization of potassium for better uptake by the plant; secretion of indoleacetic acid (IAA), the most important phytohormone for plant growth and development; and antimicrobial activity of actinomycetes – for suppression of development of pathogenic microogranisms that reduce the quality of agricultural products. The manuscript contains interesting data, however, the level of scientific presentation is insufficient. It requires substantial revision to be accepted for publication.

The remarks are as follows:

  1. The combined presentation of both biomass accumulation and pH changes on Fig. 2 is irrelevant. Either the discussion related to pH needs to be moved to section 3.1, where this diagram is located, or the biomass accumulation and pH change data need to be split into separate figures. In addition, it may be worth developing a less cumbersome way to illustrate the data obtained. Biomass accumulation plots for all strains should be combined into a single plot, and so should pH change plots, with the plots moved to the appropriate sections of Discussion.
  2. In caption to Fig. 3, there is no description of symbols A-E. Accumulation of biomass would be more clearly shown using a second column on the diagram.
  3. 4. Indole Acetic Acid (IAA) production. – remove the dot after the title
  4. Captions with names of the test strains should be added to Fig. 6, and it should be moved to Supporting Information, because it duplicates the data in Table 1.
  5. Table 2. The scoring for the studied strains are very arbitrary. Every strain is rated on a scale of 1 to 10 on completely different parameters, calculation of the “final score” does not apply any weights to various parameters. For example, assigning equal importance to antimicrobial activity and the ability to produce siderophores is very questionable in the context of biocontrol applications, as is equal importance of other mentioned parameters. A serious justification of the scoring approach with references to the methodology being used needs to be added, or this section needs to be reworked.
  6. The list of references is sloppy and needs to be thoroughly rechecked.

Author Response

Point 1 : The combined presentation of both biomass accumulation and pH changes on Fig. 2 is irrelevant. Either the discussion related to pH needs to be moved to section 3.1, where this diagram is located, or the biomass accumulation and pH change data need to be split into separate figures. In addition, it may be worth developing a less cumbersome way to illustrate the data obtained. Biomass accumulation plots for all strains should be combined into a single plot, and so should pH change plots, with the plots moved to the appropriate sections of Discussion.

Response 1 : according to the reviewer’s suggestion the biomass accumulation and pH change data were separated into two figures (Fig. 2 for Biomass and Fig. 4 for pH evolution).

Point 2 : In caption to Fig. 3, there is no description of symbols A-E. Accumulation of biomass would be more clearly shown using a second column on the diagram.

Response 2 : Corrected

Point 3 : 4. Indole Acetic Acid (IAA) production. – remove the dot after the title

Response 3 : Done

Point 4 : Captions with names of the test strains should be added to Fig. 6, and it should be moved to Supporting Information, because it duplicates the data in Table 1.

Response 4 : Corrected

Point 5 : Table 2. The scoring for the studied strains are very arbitrary. Every strain is rated on a scale of 1 to 10 on completely different parameters, calculation of the “final score” does not apply any weights to various parameters. For example, assigning equal importance to antimicrobial activity and the ability to produce siderophores is very questionable in the context of biocontrol applications, as is equal importance of other mentioned parameters. A serious justification of the scoring approach with references to the methodology being used needs to be added, or this section needs to be reworked.

Response 5 :

We agree that the non-weighted scoring approach was poorly justified in light of the central aim of the project: to identify the most promising strains for combined biofertilizer and biocontrol applications. To approach this aim, the rating of phosphate solubilisation, potassium solubilisation, indole acetic acid production and suppression of indigenous Fusarium strains were doubled (highest activity strain that was rated by 10 => value 20). Table 2 was revised and the weighted scoring was also described in the Materials and methods section.

Materials and methods, line 231

To select the most promising streptomycete isolates for combined biofertilizer and biocontrol applications, their relative activities in rock phosphate, tricalcium phosphate and potassium solubilization, antifungal activity against Fusarium, antibacterial activity, production of IAA and siderophores were cross-compared between the strains and used to grade them from 1 (lowest) to 10 (highest activity). In order to follow the aim of selecting for combined biofertilizer and biocontrol agents, the rating values for rock phosphate and potassium solubilisation, IAA production and Fusarium inhibition, were weighted by 2 (doubled).

Point 6 : The list of references is sloppy and needs to be thoroughly rechecked.

Response 6 : We regret the multiple mistakes that occurred in the References section. The section has been carefully edited.

Reviewer 2 Report

The work is devoted to the development of methods for sustainable and environmentally friendly agriculture. The objects of the study were 10 strains of actinomycetes isolated from the rhizosphere of sugar beet. The specific task of the authors was to analyze the ability of these actinomycetes to dissolve the potassium-containing mineral orthoclase, which was the only source of potassium in the nutrient medium. The ability of actinomycetes to synthesize Indole Acetic Acid and antimicrobial substances was also analyzed. As a result, the most promising strains for field trials were selected.

The disadvantages of this work are shown in the table below. They can be overcome by the authors as a result of careful reading and editing of the text.

I would like to note that when re-mentioning species names (binary nomenclature), the genus designation should be abbreviated. This rule is violated throughout the text.

Special attention is drawn to the discrepancy between the data of figure 3 and the text. This needs to be corrected, after which the work can be published.

Line

Original text

Change please on

2

 PGP

Plant growth-promotion?

Give the full name, there is no decryption anywhere in the text.

22

PGPR

Plant growth-promoting rhizobacteria?

Give the full title once in the text.

37-38

 Fusarium equiseti and two Fusarium fujikuroi

 Fusarium equiseti and two F. fujikuroi

54-55

nitrogen (N), phosphorus and potassium (K).

nitrogen (N), phosphorus (Р) and potassium (K).

90-91

From the 10,000 antimicrobial molecules produced by microorganisms, over than 50 % were isolated from Actinomycetes [40].

Work [40] is devoted to streptomycetes. In actinomycetes, about 10,000 antimicrobial compounds was known. See for example Berdy, 2005.  doi: 10.1038/ja.2005.1.

133

L tryptophan

L-tryptophan

159-160

on potato dextrose agar medium (PDA).

Provide the composition or reference.

185

and 50ng of DNA

and 50 ng of DNA

204

of the 10 selected strains [54] against the Gram negative bacteria: Escherichia coli

of the 10 selected strains [54] against the Gram negative bacteria: Escherichia coli

224

culture and S is radial growth of the fungal in the presence of Actinomycete

culture and S is radial growth of the fungus (?) in the presence of Actinomycete

248

grown in AMM+ orthoclase

grown in AMM + orthoclase

256

greatly varied from strain to strain and ranged from 3.8 mg/L to 216.6 mg/L.

Figure 3 shows other values.

253-280

Please check the text and its compliance with figure 3.

291-293

AZ, AYD, DE2 and BYC released a highest amount of K from insoluble orthoclase (216.6, 155.6 and 125.4 mg/L, respectively).

Please clarify.

315

CHAMP1, CHAMP2 and CHAMP3) belongs to the genus Fusarium. Macroscopically,

CHAMP1, CHAMP2 and CHAMP3) belongs to the genus Fusarium. Macroscopically,

332

Fusarium fujikuroi and the isolate CHAMP2 was related to Fusarium equiseti.

F. fujikuroi and the isolate CHAMP2 was related to F.equiseti.

350-353

Mark clearly on the figure a-e and in the caption A, B.

354-355

None of the 10 tested Streptomyces sp. showed an activity against the Gram positive bacteria Bacillus subtilis, Streptococcus sp. or Pseudomonas aeruginosa.

Change the phrase, for example, add "or Gram-negative bacteria Pseudomonas aeruginosa”.

356

against Staphylococus aureus.

against Staphylococcus aureus.

392-393

source. Interestingly, in the broth AMM, the amount of soluble K ranged from 3.8 mg/L to 216.6 mg/L after 5 days of cultivation.

Does not match Figure 3.

Author Response

Point 1 : The disadvantages of this work are shown in the table below. They can be overcome by the authors as a result of careful reading and editing of the text.

Response 1 : Correction done in the text

Point 2 : I would like to note that when re-mentioning species names (binary nomenclature), the genus designation should be abbreviated. This rule is violated throughout the text.

Response 2 : Correction done in the text

Point 3 : Special attention is drawn to the discrepancy between the data of figure 3 and the text. This needs to be corrected, after which the work can be published.

Response 3 : Correction done in the Figure 3.

Line

Original text

Change please on

2

 PGP

Plant growth-promotion?

Give the full name, there is no decryption anywhere in the text.

Correction Done in the text: line 2

22

PGPR

Plant growth-promoting rhizobacteria?

Give the full title once in the text.

Correction done in the text: line 22

37-38

 Fusarium equiseti and two Fusarium fujikuroi

 Fusarium equiseti and two F. fujikuroi

Done

54-55

nitrogen (N), phosphorus and potassium (K).

nitrogen (N), phosphorus (Р) and potassium (K).

Done

90-91

From the 10,000 antimicrobial molecules produced by microorganisms, over than 50 % were isolated from Actinomycetes [40].

Work [40] is devoted to streptomycetes. In actinomycetes, about 10,000 antimicrobial compounds was known. See for example Berdy, 2005.  doi: 10.1038/ja.2005.1.

Correction done in the text: line 90-91 : From the 10,000 antimicrobial molecules produced by Actinomycetes, over than 50 % were isolated from Streptomycetes [40].

133

L tryptophan

L-tryptophan

Done line 133

159-160

on potato dextrose agar medium (PDA).

Provide the composition or reference.

Composition provided, line 159

185

and 50ng of DNA

and 50 ng of DNA

Done, line 186

204

of the 10 selected strains [54] against the Gram negative bacteria: Escherichia coli

of the 10 selected strains [54] against the Gram negative bacteria: Escherichia coli

Correction done in the text: line 204

224

culture and S is radial growth of the fungal in the presence of Actinomycete

culture and S is radial growth of the fungus (?) in the presence of Actinomycete

Correction done in the text: line 222-226

248

grown in AMM+ orthoclase

grown in AMM + orthoclase

Done

256

greatly varied from strain to strain and ranged from 3.8 mg/L to 216.6 mg/L.

Figure 3 shows other values.

Sorry, the Figure 3 had a too low value for the strain AZ. It has been changed.

253-280

Please check the text and its compliance with figure 3.

291-293

AZ, AYD, DE2 and BYC released a highest amount of K from insoluble orthoclase (216.6, 155.6 and 125.4 mg/L, respectively).

Please clarify.

Correction done in the text: line 280

315

CHAMP1, CHAMP2 and CHAMP3) belongs to the genus FusariumMacroscopically,

CHAMP1, CHAMP2 and CHAMP3) belongs to the genus Fusarium. Macroscopically,

Correction done in the text: line 327

332

Fusarium fujikuroi and the isolate CHAMP2 was related to Fusarium equiseti.

F. fujikuroi and the isolate CHAMP2 was related to F.equiseti.

Correction done in the text: line 343

350-353

Mark clearly on the figure a-e and in the caption A, B.

Correction done in the figure 7

354-355

None of the 10 tested Streptomyces sp. showed an activity against the Gram positive bacteria Bacillus subtilisStreptococcus sp. or Pseudomonas aeruginosa.

Change the phrase, for example, add "or Gram-negative bacteria Pseudomonas aeruginosa”.

Correction done in the text: line 377

356

against Staphylococus aureus.

against Staphylococcus aureus.

Done

392-393

source. Interestingly, in the broth AMM, the amount of soluble K ranged from 3.8 mg/L to 216.6 mg/L after 5 days of cultivation.

Does not match Figure 3.

Sorry, Figure 3 value of strain AZ was wrong. It has been changed.

Round 2

Reviewer 1 Report

Authors addressed all the suggested issues. In present form the manuscript is suitable for publishing in Microorganisms.